# Co-Inhibition of PARP and STAT3 as a Promising Approach for Triple-Negative Breast Cancer

**DOI:** 10.3390/biom15071035

**Published:** 2025-07-17

**Authors:** Changyou Shi, Li Pan, Satomi Amano, Mei-Yi Wu, Chenglong Li, Jiayuh Lin

**Affiliations:** 1Department of Biochemistry and Molecular Biology, University of Maryland School of Medicine, Baltimore, MD 21201, USA; changyou.shi@som.umaryland.edu (C.S.); li.pan@som.umaryland.edu (L.P.); samano@som.umaryland.edu (S.A.); mwu@som.umaryland.edu (M.-Y.W.); 2Department of Medicinal Chemistry, College of Pharmacy, University of Florida, Gainesville, FL 32611, USA; lic@ufl.edu

**Keywords:** triple-negative breast cancer (TNBC), *BRCA*, PARP inhibitor, STAT3 inhibition, IL-6/STAT3 signaling

## Abstract

Triple-negative breast cancer (TNBC) is a highly aggressive subtype known for its rapid metastatic potential. Despite its severity, treatment options for TNBC remain limited. Olaparib, an FDA-approved PARP inhibitor, has been used to treat germline *BRCA*-mutated TNBC in both metastatic and high-risk early-stage settings. However, acquired resistance to PARP inhibitors and their limited applicability in non-*BRCA* TNBCs are now two major growing clinical problems. Activation of the IL-6/STAT3 signaling cascade has been implicated in therapeutic resistance. In this study, we evaluated the combined effects of the PARP inhibitor olaparib and the STAT3 inhibitor LLL12B in human TNBC cell lines with both *BRCA* mutations and wild-type *BRCA* status. Our results demonstrate that the PARP inhibitor olaparib can induce increased interleukin-6 (IL-6) in TNBC cells, with ELISA showing a 2- to 39-fold increase across five cell lines. MTT assays revealed that knocking down or inhibiting STAT3, a key downstream effector of the IL-6/GP130 pathway, sensitizes TNBC cells to olaparib. Treatment with either olaparib or LLL12B alone reduced TNBC cell viability, migration, and invasion. Notably, their combined administration produced a markedly enhanced inhibitory effect compared to individual treatments, regardless of *BRCA* mutation status. These findings highlight the potential of dual PARP and STAT3 inhibition as a novel targeted therapeutic strategy for both *BRCA*-mutant and *BRCA*-proficient TNBC.

## 1. Introduction

Triple-negative breast cancer (TNBC) is highly metastatic with few effective therapeutic options and is a major cause of death among breast cancer patients [1]. Therefore, there is a critical medical need to find novel approaches for TNBC therapy to reduce patient mortality. Approximately 15–20% of TNBC harbor *BRCA* mutations, which render these cells lacking the ability to accurately repair lethal DNA double-strand breaks (DSBs) via the homologous recombination (HR) pathway during the S/G2 phase of the cell cycle [2]. Poly (ADP-ribose) polymerase (PARP) repairs single-strand breaks (SSBs) and PARP inhibitors lead to persistence of SSBs that are converted to DSBs during DNA replication [3]. In *BRCA*-mutant breast cancer cells, DSBs induced by PARP inhibition are not repaired efficiently, resulting in cell death by a mechanism known as synthetic lethality [4]. While *BRCA*-mutant TNBC may respond to single-agent PARP inhibitor (PARPi) initially, it is not durable, leading to disease progression and resistance to therapy [5]. Importantly, most *BRCA*-proficient TNBCs exhibit limited responsiveness to PARP inhibitor therapy due to intact HR repair, which efficiently mitigates PARP-induced DNA damage and reduces treatment efficacy [6]. Recent reports suggest that combining PARP inhibitors (PARPi) with select drugs may be effective in BRCA-proficient cancer cells, supporting the idea that novel PARPi combination approaches may fulfill the unmet therapeutic need in TNBC [7].

Interleukin-6 (IL-6)/phosphorylated STAT3 levels are elevated in the majority of TNBC specimens and lymph node metastases and persistent STAT3 phosphorylation is enriched in TNBC [8]. Once activated, STAT3 moves into the nucleus [9,10], where it modulates the transcription of genes that govern diverse cellular functions, such as those regulating cell proliferation (e.g., Cyclin D1, c-Myc), apoptosis resistance (e.g., Survivin, Bcl-xL, Bcl-2), blood vessel formation (e.g., VEGF), and cellular motility and matrix degradation (e.g., MMP-2) [11,12,13]. These processes collectively confer resistance to anti-cancer drugs, promoting cell survival and metastasis. Due to its central involvement in tumor initiation and advancement, the IL-6/STAT3 pathway is increasingly recognized as a promising focus for small-molecule drug development. Several therapeutic agents have been developed to inhibit the IL-6/STAT3 pathway, targeting upstream components such as IL-6, its receptor, or associated kinases like Janus kinases (JAKs). These include biologics, such as Tocilizumab [14], an IL-6 receptor monoclonal antibody, and small molecules, such as Ruxolitinib [15], a JAK1/2 inhibitor. While these agents have demonstrated efficacy in reducing STAT3 activation, they are limited by their indirect approach. Targeting upstream components can lead to off-target effects and unintended suppression of other pathways critical for normal physiological functions, such as immune regulation and hematopoiesis [16,17]. Moreover, alternative routes such as cytokine and growth factor receptors, oncogenic kinases (e.g., SRC, BCR-ABL), or non-canonical pathways involving mitochondrial STAT3 and unphosphorylated STAT3 can allow cancer cells to bypass upstream inhibition [18]. 

Therefore, the development of novel drugs that specifically target STAT3 itself is essential for improving treatment outcomes. Various STAT3 inhibitors with distinct mechanisms have been developed, with some currently undergoing clinical trials. Despite ongoing research, STAT3-targeting agents have yet to gain FDA approval for clinical use in cancer therapy [19]. Therefore, the pursuit of highly specific and potent compounds targeting STAT3 activity remains a critical goal for potential cancer prevention and therapy.

LLL12B is a newly developed, orally administered small molecule with inhibitory activity targeting STAT3. LLL12B interacts specifically with the Tyr705 phosphorylation site on STAT3, thereby preventing its dimerization, nuclear translocation, and subsequent transcriptional activity [20,21]. Preclinical studies have demonstrated that LLL12B effectively inhibits tumor cell growth, migration, and invasion across various cancer types including TNBC [22], medulloblastoma [20], ovarian cancer [23], etc., highlighting its potential as a novel therapeutic agent for STAT3-driven malignancies. In addition to LLL12B, other direct STAT3 inhibitors such as TTI-101 (C188-9) [24], LYW-6 [25], and dual-site inhibitors like compound 4c [26] have shown promising antitumor effects in early-phase clinical or preclinical studies, further underscoring the therapeutic relevance of targeting STAT3 across cancer types.

In the present study, we tested the combination treatment of LLL12B and olaparib, and the results showed that co-targeting PARP and STAT3 can further suppress the viability, migration, and invasion of both *BRCA*-mutant and *BRCA*-proficient TNBC cells in vitro. These findings highlight the potential of this combination as an innovative and enhanced treatment strategy for TNBC compared to PARP inhibitor monotherapy.

## 2. Materials and Methods

### 2.1. Reagents

LLL12B was developed by Dr. Chenglong Li’s research group at the College of Pharmacy, University of Florida. Olaparib (catalog no. HY-16106) was obtained from MedChemExpress LLC, located in Monmouth Junction, NJ, USA. Sterile dimethyl sulfoxide (DMSO) was used to prepare 20 mM stock solutions of both compounds, which were then stored at −20 °C. 3- (4,5-Dimethylthiazol-2-yl)-2,5-diphenyltetrazolium bromide (MTT) was obtained from MilliporeSigma (Merck KGaA), and N,N-dimethylformamide (DMF), cat. no. 047390, was purchased from Thermo Fisher Scientific, Inc. (Waltham, MA, USA).

Drug concentrations used in each experimental assay (e.g., MTT viability assay, Western blot, migration, and invasion assays) were optimized based on pilot experiments and literature precedent. The sensitivity of TNBC cells to LLL12B and olaparib varied depending on factors such as cell line characteristics, seeding densities, growth kinetics, and the type of culture system employed (e.g., 96-well plates for viability assays, 10 cm plates for protein analysis, or transwell chambers for invasion assays). Therefore, appropriate drug concentrations were selected for each assay to ensure reliable measurement of biological effects while minimizing excessive cytotoxicity that could confound interpretation. In all assays, the final concentration of DMSO was kept consistent across experimental and control groups and did not exceed 0.1%, ensuring appropriate vehicle control comparability.

### 2.2. Cell Culture

Human TNBC cell lines used in this study included *BRCA1*-mutant (MDA-MB-436, HCC1937, and SUM149) and *BRCA1* wild-type [MDA-MB-231 and preferential bone metastatic subline MDA-MB-231 bone-metastasis (MDA-MB-231 BoneM)]. MDA-MB-436, MDA-MB-231, and MDA-MB-231 BoneM cells were maintained in Dulbecco’s Modified Eagle Medium (DMEM; Corning, NY, USA) supplemented with 10% fetal bovine serum (FBS; Sigma-Aldrich, St. Louis, MO, USA) and 1% penicillin-streptomycin (P/S; Sigma-Aldrich). HCC1937 and SUM149 cells were cultured in Roswell Park Memorial Institute (RPMI) 1640 Medium (Corning Inc., Corning, NY, USA) supplemented with 10% FBS and 1% P/S. All cell lines were maintained at 37 °C in a humidified atmosphere with 5% CO_2_. All cell lines were authenticated by short tandem repeat (STR) profiling and routinely tested for mycoplasma contamination using a PCR-based detection kit. Only mycoplasma-free cells were used for experiments.

### 2.3. IL-6 Determination

TNBC cells were seeded in 24-well plates at 50% confluency and incubated with either the drug or DMSO in serum-free medium for 3 days. The concentration of olaparib used varied across cell lines and was selected based on preliminary dose–response viability assays, as different TNBC cell lines exhibited differential sensitivity to the drug. Specifically, doses were chosen to achieve comparable partial inhibition (~30–50% reduction in viability) to allow for consistent comparison of IL-6 induction under sub-lethal stress conditions. Following treatment, culture supernatants were harvested from each well, and IL-6 concentrations were determined using the Quantikine ELISA kit (cat. no. D6050; R&D Systems, Inc., Minneapolis, MN, USA) following the supplier’s protocol. Absorbance values (Optical density) were then recorded at 450 nm and determined using an 800 TS microplate reader (BioTek, Winooski, VT, USA). The assay has a minimum detectable dose (sensitivity) of 0.70 pg/mL and a linear detection range of 3.1–300 pg/mL.

### 2.4. SiRNA-Mediated Knockdown of STAT3

To suppress STAT3 expression, MDA-MB-436, HCC1937, SUM149, MDA-MB-231, and MDA-MB-231 BoneM cells were transfected with either STAT3-targeting siRNA or non-targeting control siRNA (Cell Signaling Technology, Danvers, MA, USA). Transfections were performed at approximately 60–80% confluency using 100 nm siRNA and Lipofectamine RNAiMAX (Invitrogen, Carlsbad, CA, USA) following the manufacturer’s protocol. Three days post-transfection, cell viability was measured via the MTT assay, and STAT3 knockdown efficiency was confirmed by Western blotting. Although densitometric quantification was attempted, in several cell lines (MDA-MB-436, SUM149, and MDA-MB-231), STAT3 protein levels were nearly or completely undetectable after siRNA treatment, preventing reliable quantification. The absence or marked reduction in the STAT3 band indicated effective gene silencing.

### 2.5. MTT Cell Viability Assay

All TNBC cell lines were plated in triplicate into 96-well plates at a density of 3000 cells per well. After overnight incubation, the cells were exposed to varying concentrations of the LLL12B and/or olaparib, or to DMSO as a vehicle control, for a duration of 72 h. Following treatment, 20 µL of MTT reagent (cat. no. 475989; Millipore Sigma, Rockville, MD, USA) was added to each well and incubated for an additional 4 h. Then, 150 µL of DMF solubilization solution was introduced, and the plates were shaken overnight in the dark to ensure complete dissolution of the formazan crystals. Absorbance was assessed by measuring absorbance at 595 nm using an 800 TS microplate reader.

### 2.6. Western Blotting

Cells were harvested 24 h after treatment, and total protein was extracted using cell lysis buffer (cat. no. 9803S; Cell Signaling Technology, Inc., Danvers, MA, USA). Protein concentrations were measured using the Pierce BCA Protein Assay Kit (Thermo Fisher Scientific, Inc.) in accordance with the manufacturer’s protocol. Equivalent amounts of protein were separated on 8% SDS-PAGE gels and transferred to PVDF membranes. The membranes were blocked with 5% non-fat milk at room temperature for 1 h, followed by overnight incubation at 4 °C with primary antibodies targeting total STAT3 (cat. no. 12640S), phosphorylated STAT3 (Y705) (cat. no. 9145S), or GAPDH (cat. no. 2118S), all at a 1:1000 dilution; Cell Signaling Technology, Inc. (Danvers, MA, USA). The following day, the membranes were incubated with an HRP-linked anti-rabbit secondary antibody (1:2000; cat. no. 7074S; Cell Signaling Technology, Inc., Danvers, MA, USA) at room temperature for 1 h. Detection of protein bands was performed using the SuperSignal™ West Femto Maximum Sensitivity Substrate (cat. no. 34094; Thermo Fisher Scientific, Inc., Waltham, MA, USA) and visualized with an Amersham Imager 680 (Cytiva, Marlborough, MA, USA).

### 2.7. Immunofluorescence

MDA-MB-231 cells were plated on glass coverslips sterilized with 70% ethanol and incubated overnight in DMEM supplemented with 10% FBS. The next day, cells were treated with 10 µM olaparib or an equivalent volume of DMSO in serum-free DMEM containing 1% P/S for 24 h. After treatment, cells were rinsed with PBS, fixed in 4% paraformaldehyde for 15 min, and permeabilized using 0.3% Triton X-100 (cat. no. 9036-19-5; Sigma-Aldrich, St. Louis, MO, USA) for an additional 15 min. Blocking was performed in PBS containing 5% BSA and 0.3% Triton X-100 for 1 h at room temperature. Cells were then incubated overnight at 4 °C with an anti-STAT3 primary antibody (Cell Signaling Technology, cat. no. 12640) diluted 1:100 in 5% BSA. The following day, samples were exposed to Alexa Fluor 488-conjugated donkey anti-rabbit secondary antibody (Invitrogen, cat. no. A-21206) at a 1:1000 dilution for 1 h at room temperature. Coverslips were mounted using Fluoromount-G with DAPI (cat. no. 00-4959-52; Invitrogen, Carlsbad, CA, USA), and fluorescence images were captured with a 40× objective on an AxioObserver 7 Motorized System (Zeiss, Jena, Germany). Imaging settings were kept consistent across all samples.

### 2.8. Wound Healing Assay

The MDA-MB-436, HCC1937, SUM149, MDA-MB-231, and MDA-MB-231 BoneM cells were plated in 6-well plates and grown to confluence. A uniform linear wound was created across the monolayer using a 200 µL pipette tip. Images of the scratched areas were captured at 0 h using an Echo Rebel microscope (ECHO; BICO), prior to treatment. Cells were subsequently exposed in triplicate to different concentrations of LLL12B, olaparib, their combination, or DMSO as a vehicle control. After incubation, images were taken again once scratch closure was nearly complete in DMSO-treated controls. Scratch areas at each time point were measured using the Echo Rebel microscope (Echo, San Diego, CA, USA) equipped with integrated image capture and analysis software, the scratch closure rate was calculated using the formula: (initial scratch area—final scratch area after incubation)/initial scratch area, and the relative migration percentage was normalized to the DMSO-treated control.

### 2.9. In Vitro Invasion Assay

Cell invasive capacity was assessed using transwell inserts pre-coated with Matrigel. The 24-well inserts (CELLTREAT Scientific Products, Pepperell, MA, USA) were layered with 1 mg/mL Matrigel (Corning Inc., Corning, NY, USA) and incubated at 37 °C for a minimum of 2 h to allow gel solidification. Roughly 50,000 cells suspended in 200 µL of serum-free DMEM or RPMI were added to each insert. Treatments were applied in triplicate using different concentrations of LLL12B, olaparib, their combination, or DMSO as a control. The lower chambers contained 500 µL of DMEM or RPMI supplemented with 10% FBS to serve as a chemoattractant.

After 20–24 h of incubation, non-invading cells were removed and the invasive cells on the bottom surface of the membrane were fixed with methanol and stained with 0.1% crystal violet. Images were captured using an Echo Rebel microscope at ×100 magnification. The number of invasive cells was quantified using ImageJ software (Version 1.54d; National Institutes of Health, Bethesda, MD, USA), and results were expressed as the average number of invasive cells. For each well, five images were captured, and each condition was tested in triplicate.

### 2.10. Statistical Analysis

All data are expressed as mean  ±  standard error (SE). Comparisons between two groups were evaluated using Student’s *t*-test, while multiple group comparisons were assessed via ANOVA where appropriate. Statistical analyses were conducted using GraphPad software. (Version 10.5; Graphpad Software, San Diego, CA, USA). Significance levels were denoted as follows: * *p*  <  0.05, ** *p*  <  0.01, *** *p*  <  0.001 and **** *p*  <  0.0001.

## 3. Results

### 3.1. Olaparib Enhances IL-6 Production in Human TNBC Cells

To assess the impact of the PARP inhibitor olaparib on IL-6 production, human TNBC cell lines MDA-MB-436, HCC1937, SUM149 and MDA-MB-231 were treated with either DMSO or olaparib for six days. Following treatment, cell culture supernatant was collected and IL-6 concentrations measured by ELISA. As shown in Figure 1, olaparib treatment significantly induced IL-6 secretion, with approximate fold increases of 10.7 in SUM149, 39.0 in MDA-MB-231, 4.5 in MDA-MB-231 BoneM, 9.9 in MDA-MB-436, and 1.9 in HCC1937 cells.

### 3.2. STAT3 Silencing Reduces TNBC Cell Viability and Promotes Sensitivity to Olaparib

Given that olaparib treatment markedly upregulated IL-6 expression in TNBC cells, we investigated whether silencing its key downstream effector, STAT3, could enhance the cells’ sensitivity to olaparib.

Western blot analysis verified successful STAT3 silencing, as evidenced by diminished levels of total STAT3 and its phosphorylated form at Y705 across all TNBC cell lines (Figure 2A). MTT assays revealed that olaparib treatment reduced cell viability in most TNBC cell lines, except for MDA-MB-436, which exhibited limited sensitivity (Figure 2B, C vs. C + O15). Second, knockdown of STAT3 using siRNA significantly decreased cell viability across all five TNBC cell lines tested, including both *BRCA*-mutant (MDA-MB-436, HCC1937, and SUM149) and *BRCA*-proficient (MDA-MB-231 and MDA-MB-231 BoneM) cells (Figure 2B, C vs. Si). Notably, STAT3 knockdown further enhanced the inhibitory effect of olaparib on cell viability compared to olaparib treatment alone in control siRNA-transfected TNBC cells, regardless of *BRCA1* mutation status (Figure 2B, C + O15 vs. Si + O15). These results suggest that STAT3 supports TNBC cell survival and that its suppression sensitizes cells to PARP inhibition.

### 3.3. Olaparib Combined with LLL12B Synergistically Inhibits the Viability of TNBC Cells

To assess potential synergistic effects of LLL12B and olaparib on TNBC cells, MDA-MB-436, HCC1937, SUM149, MDA-MB-231 and MDA-MB-231 BoneM cells were exposed to each agent alone or in combination for 72 h. As illustrated in Figure 3, treatment with either LLL12B or olaparib alone reduced cell viability in all TNBC cell lines, except for MDA-MB-436, which showed no significant response to olaparib monotherapy compared to the DMSO control. However, the combination of both drugs led to a markedly stronger reduction in cell viability than treatment with either agent alone across all human TNBC cell lines, indicating a synergistic effect in suppressing cell viability.

### 3.4. Co-Treatment with Olaparib and LLL12B Suppresses Olaparib-Induced STAT3 Activation in TNBC Cells

To elucidate the mechanism underlying the inhibitory effects of the combined treatment with olaparib and LLL12B in TNBC cells, we conducted Western blot analysis. As shown in Figure 4A, olaparib treatment alone led to an increase in STAT3 phosphorylation compared to the DMSO-treated control. However, co-treatment with LLL12B markedly suppressed this olaparib-induced phosphorylation of STAT3 (Figure 4A and Appendix A). To further confirm whether STAT3 translocates to the nucleus following olaparib treatment, we performed immunofluorescence staining to assess its subcellular localization in MDA-MB-231 cells. As shown in Figure 4B, olaparib-treated cells exhibited enhanced nuclear localization of STAT3 compared to DMSO controls, suggesting that olaparib-induced IL-6 upregulation promotes STAT3 activation and nuclear translocation.

### 3.5. LLL12B Plus Olaparib Combination Treatment Inhibits TNBC Cell Migration

Would-healing assays were performed to evaluate the potential synergistic inhibitory effect of LLL12B combined with olaparib on TNBC cell migration. Representative images and quantitative analysis are presented in Figure 5. Monotherapy with either LLL12B or olaparib reduced cell migration in most TNBC cell lines; however, SUM149 showed no significant response to LLL12B alone, and MDA-MB-231 exhibited no significant reduction in migration with olaparib treatment compared to DMSO control. However, the combination of both drugs exhibited a much more significantly inhibitory effect on migration compared to either monotherapy. This enhanced suppression was statistically significant when comparing the combination treatment to each single drug across all TNBC cell lines, indicating that the combination therapy may be more effective in limiting TNBC cell migration.

### 3.6. The Combination of LLL12B and Olaparib Suppresses Invasion of TNBC Cells

The ability of tumor cells to invade surrounding tissues is a key feature driving metastasis. To investigate the potential of LLL12B and/or olaparib in inhibiting invasion, a cell invasion assay was conducted using MDA-MB-436, HCC1937, SUM149 and MDA-MB-231 TNBC cell lines. Both LLL12B and olaparib monotherapies reduced tumor cell invasion across most TNBC cell lines (Figure 6A–E), with quantification shown in Figure 6F. However, olaparib treatment did not reduce invasion in HCC1937 or MDA-MB-231 BoneM cells. Notably, the combination treatment produced a more pronounced inhibition of invasion, ranging from 70% to 93% relative to the DMSO control in all TNBC cell lines, except for MDA-MB-231 BoneM, which showed modest inhibition of 40%. These results highlight the enhanced efficacy of dual therapy in suppressing the invasive potential in TNBC cells.

A schematic model summarizing the proposed mechanism by which olaparib and LLL12B cooperatively suppress TNBC progression is presented in Figure 7.

## 4. Discussion

TNBC accounts for 15–20% of all breast cancer cases worldwide [27], with patients exhibiting an earlier age of onset, lower five-year survival rates, and higher incidences of recurrence and metastasis to vital organs such as the brain, lung, and liver [28]. Because TNBC lacks estrogen receptor (ER), progesterone receptor (PR), and HER2 expression, conventional hormone therapies and HER2-targeted agents are ineffective. Therefore, there remains a critical need for novel therapeutic strategies, especially for managing metastatic TNBC [29].

PARPs are crucial enzymes involved in preserving genomic integrity by mediating DNA repair, especially via the base excision repair (BER) mechanism. Among them, PARP-1 has been the subject of the most extensive research, acting as a sensor for SSBs and initiating repair by recruiting the necessary repair proteins. When left unrepaired, SSBs can progress to DSBs during replication [30,31]. In normal cells, homologous recombination, a precise repair mechanism dependent on *BRCA1* and *BRCA2*, resolves these breaks [32]. However, in cancers with *BRCA1/2* mutations, homologous recombination is impaired, forcing cells to depend on less accurate and alternative DNA repair mechanisms, such as PARP-mediated repair, which makes them particularly susceptible to PARP inhibition [33].

Olaparib, a PARP inhibitor, has received FDA approval for the treatment and maintenance of several cancers, particularly those with a germline *BRCA1* or *BRCA2* mutation, including HER2-negative metastatic breast cancer [34,35,36,37]. Despite its efficacy in treating *BRCA*-mutant cancers, olaparib has several limitations that hinder its broader clinical application. One major challenge is acquired resistance, where cancer cells restore homologous recombination repair through *BRCA1/2* reversion mutations, loss of 53BP1, or increased drug efflux, reducing olaparib’s effectiveness [38]. Numerous mechanisms underlying resistance to PARP inhibitors have been described, among which activation of the IL-6/STAT3 signaling pathway is notable [39]. Researchers discovered that elevated A2B receptor expression in ovarian cancer cells senses adenosine signals, triggering the IL-6-STAT3 signaling cascade, which contributes to tumor cell survival, proliferation, migration, and reduced sensitivity to the PARP inhibitor olaparib [39]. Our study also shows a significant increase in IL-6 levels with olaparib treatment in both *BRCA*-mutant and *BRCA*-proficient TNBCs (Figure 1), consistent with findings in ovarian cancer. Notably, IL-6/STAT3 signaling has also been shown to support DNA damage repair mechanisms, including HR and non-homologous end joining, potentially contributing to resistance against DNA-damaging agents such as PARP inhibitors [40,41]. More importantly, STAT3 knockdown significantly enhances the sensitivity of both *BRCA*-mutant and *BRCA*-proficient TNBC cells to olaparib (Figure 2), suggesting that STAT3 may play a broader role in mediating cellular resistance mechanisms beyond HR deficiency. One plausible explanation is that STAT3 supports alternative survival pathways, including anti-apoptotic and inflammatory signaling [42], which may buffer the cytotoxic effects of PARP inhibition. Additionally, prior studies have suggested that STAT3 may influence the expression of DNA repair genes and modulate the tumor microenvironment to promote therapy resistance [41,43]. Thus, dual inhibition of PARP and STAT3 may impair compensatory survival signals and sensitize TNBC cells to DNA damage-induced apoptosis, regardless of *BRCA* mutation status.

LLL12B is a novel, orally bioavailable small molecule that selectively inhibits IL-6-mediated STAT3 activation [22]. Limited research from our lab has demonstrated its antitumor activity, either used alone or alongside agents like paclitaxel or cisplatin across various cancer types, including TNBC [20,22,23]. In this study, we evaluated the combined effects of LLL12B and olaparib on human TNBC cell lines in vitro. MTT assay results demonstrated that, although both drugs individually reduced cell viability, their combination produced a synergistic effect, resulting in more pronounced inhibition than either agent alone. Similarly to STAT3 knockdown, LLL12B treatment also enhances the sensitivity of TNBC cells to olaparib (Figure 3). These results support that STAT3 inhibition can sensitize TNBC cells to olaparib treatment. This synergistic effect is consistent with findings in ovarian and palbociclib-resistant, ER-positive breast cancers, where STAT3 inhibitors (e.g., C188-9, napabucasin, and TTI-101) combined with olaparib or other PARP inhibitors have shown enhanced anti-cancer efficacy, including increased apoptosis and reduced tumor growth [39,44]. Given that migration and invasion are key components of the metastatic cascade, we further assessed how drug treatments influence these processes. Our results support that the combined use of olaparib and LLL12B may effectively impair the metastatic potential of TNBC cells. The observed synergistic effect may be attributed to a more effective inhibition of the IL-6/STAT3 signaling pathway, which plays a critical role in promoting metastasis-related processes. STAT3 activation facilitates epithelial-to-mesenchymal transition (EMT) by modulating the expression of key transcriptional regulators and adhesion molecules, enhancing mesenchymal markers while reducing epithelial characteristics [45]. In addition, STAT3 contributes to tumor cell invasion by upregulating matrix-degrading enzymes such as MMP-2 and MMP-9 [46]. Therefore, the combined use of a PARP inhibitor with LLL12B, a STAT3 pathway inhibitor, may concurrently impair EMT dynamics and degrade matrix remodeling activity, producing a more pronounced reduction in cell motility and invasiveness than either agent alone. However, since these findings are based on in vitro assays, further in vivo studies are needed to validate the anti-metastatic efficacy of this combination and to better understand its therapeutic potential in a physiological context.

## 5. Conclusions

As illustrated in our model (Figure 7), the induction of IL-6/STAT3 signaling by olaparib may contribute to TNBC progression and resistance, which can be counteracted by STAT3 inhibition through LLL12B. The combination of olaparib and LLL12B not only enhances the sensitivity of *BRCA*-mutant TNBC to olaparib but also expands its application to *BRCA*-proficient TNBC patients. The olaparib and LLL12B combination treatment may also have the potential to offer a new targeted treatment strategy for TNBC. Further research is needed to evaluate this drug combination in vivo as a potentially targeted anti-cancer therapy in TNBC.

## Figures and Tables

**Figure 1 biomolecules-15-01035-f001:**
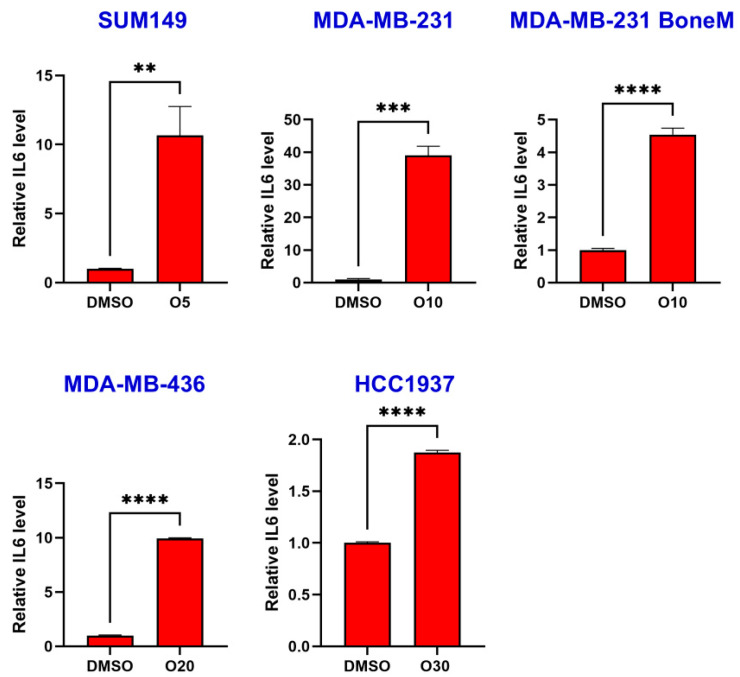
Olaparib induced IL-6 secretion in TNBC cells. SUM149, MDA-MB-231, MDA-MB-231 BoneM, MDA-MB-436, and HCC1937 cells were treated with increasing concentrations of olaparib (5 µM, 10 µM, 20 µM, and 30 µM; labeled as O5 → O30) for 6 days and IL-6 secretion was determined by ELISA assay. Data are presented as the mean ± standard error. ** *p* < 0.01, *** *p* < 0.001, **** *p* < 0.0001.

**Figure 2 biomolecules-15-01035-f002:**
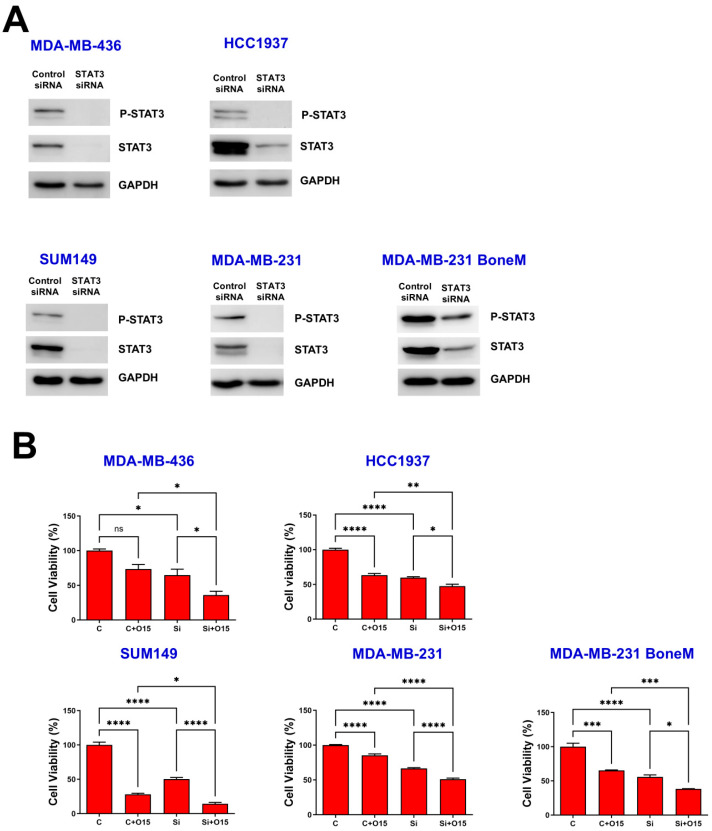
Knockdown of STAT3 inhibited TNBC cell viability. MDA-MB-436, HCC1937, SUM149, MDA-MB-231 and MDA-MB-231 BoneM cells were exposed to either control or STAT3-targeting siRNA for 72 h, Western blotting was used to examine the levels of phosphorylated STAT3 (Y705) and total STAT3 (**A**), with GAPDH included as a loading control. Cell viability was assessed using the MTT assay (**B**). Values are shown as mean ± SE. Statistical significance is indicated as follows: ns, no significant; * *p* < 0.05; ** *p* < 0.01; *** *p* < 0.001; **** *p* < 0.0001. C, control siRNA; Si, STAT3 siRNA; O15, 15 µM olaparib. Original Western blot images can be found in the Appendix A.

**Figure 3 biomolecules-15-01035-f003:**
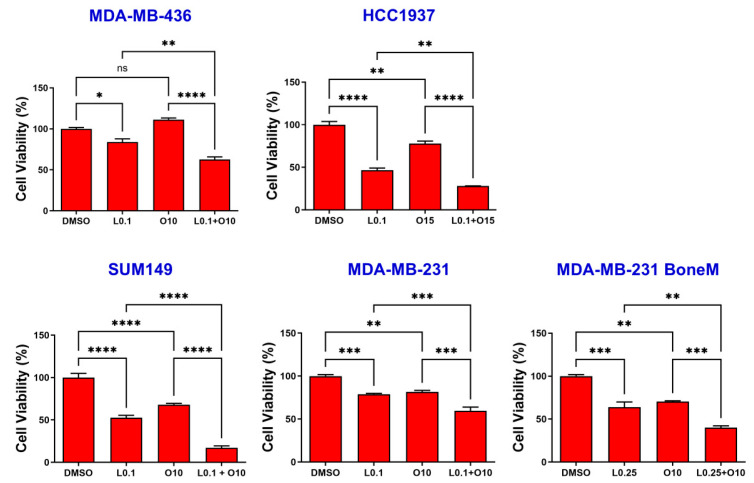
Effects of single and combined olaparib and LLL12B on TNBC cell viability. TNBC cell lines MDA-MB-436, HCC1937, SUM149, MDA-MB-231, and MDA-MB-231 BoneM were treated with varying concentrations of olaparib and/or LLL12B for 3 days. Two drug combination exhibited a synergistic effect, and inhibited cell viability more significantly than either single drug. Drug concentrations were optimized for each cell line to balance efficacy and minimize cytotoxicity. Values are shown as mean ± SE. Statistical significance is indicated as follows: ns, no significant; * *p* < 0.05; ** *p* < 0.01; *** *p* < 0.001; **** *p* < 0.0001. L0.1, 0.1 µM of LLL12B; L0.25, 0.25 µM of LLL12B; O10, 10 µM of olaparib; O15, 15 µM of Olaparib.

**Figure 4 biomolecules-15-01035-f004:**
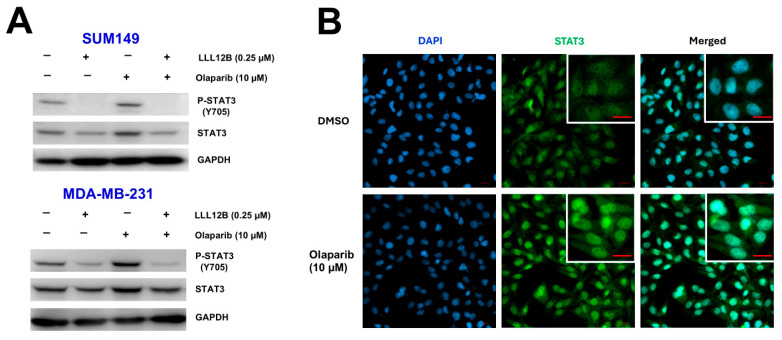
Inhibition of STAT3 activation by LLL12B in olaparib-treated TNBC cells. (**A**) Western blot analysis of phosphorylated STAT3 (P-STAT3) and total STAT3 in SUM149 and MDA-MB-231 cells following treatment with DMSO, olaparib, LLL12B, or the combination of olaparib and LLL12B for 24 h. GAPDH was used as a loading control. (**B**) Immunofluorescence staining of MDA-MB-231 cells treated with DMSO or olaparib (10 μM) for 24 h. Cells were stained with STAT3 primary antibody followed by Alexa Fluor 488-conjugated secondary antibody (green). Nuclei were counterstained with DAPI (blue). Increased nuclear localization of STAT3 was observed in olaparib-treated cells. Scale bar: 20 μm. Original Western blot images can be found in the Appendix A.

**Figure 5 biomolecules-15-01035-f005:**
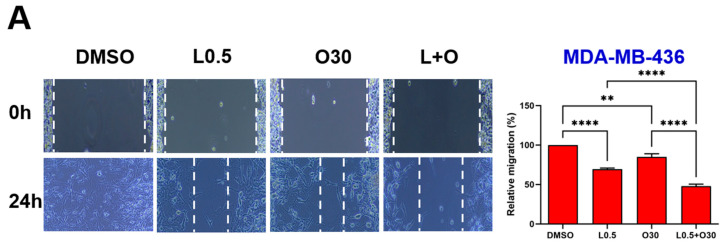
Effects of single and combined olaparib and LLL12B on TNBC cell migration. A wound healing assay was performed in (**A**) MDA-MB-436, (**B**) HCC1937, (**C**) SUM149, (**D**) MDA-MB-231, and (**E**) MDA-MB-231 BoneM cells. Cells were treated with olaparib, LLL12B, or their combination, and wound closure was monitored to assess migration. Combination treatment significantly inhibited migration compared to either single agent or DMSO. Doses were selected based on preliminary optimization for each assay system. Images were captured at a magnification of ×10. Data are shown as mean ± SE. Statistical significance is indicated as follows: ns, no significant; * *p* < 0.05; ** *p* < 0.01; *** *p* < 0.001; **** *p* < 0.0001. L0.25, 0.25 µM of LLL12B; L0.5, 0.5 µM of LLL12B; L1, 1 µM of LLL12B; O15, 15 µM of olaparib; O30, 30 µM of Olaparib.

**Figure 6 biomolecules-15-01035-f006:**
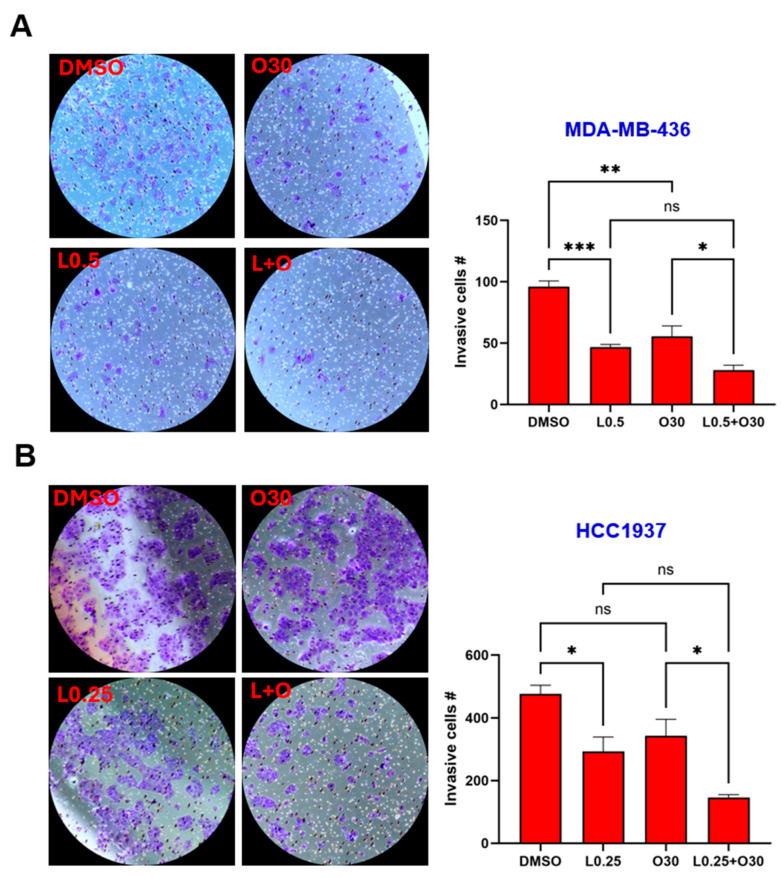
Inhibitory effects of Olaparib, LLL12B, and their combination on cell invasion. Invasion assays were performed using Matrigel-coated transwell chambers for (**A**) MDA-MB-436, (**B**) HCC1937, (**C**) SUM149, (**D**) MDA-MB-231, and (**E**) MDA-MB-231 BoneM cells treated with olaparib, LLL12B, or their combination. (**F**) Quantification of invasive inhibition percentages relative to DMSO controls in TNBC cells. Combination treatment resulted in a stronger suppression of cell invasion compared to monotherapy across most cell lines. Images were captured at a magnification of ×10. Drug concentrations were optimized to capture measurable inhibition while avoiding excessive cytotoxicity. Data are shown as mean ± SE. Statistical significance is indicated as follows: ns, no significant; * *p* < 0.05; ** *p* < 0.01; *** *p* < 0.001; **** *p* < 0.0001. L0.25, 0.25 µM of LLL12B; L0.5, 0.5 µM of LLL12B; L1, 1 µM of LLL12B; O30, 30 µM of olaparib; O35, 35 µM of olaparib.

**Figure 7 biomolecules-15-01035-f007:**
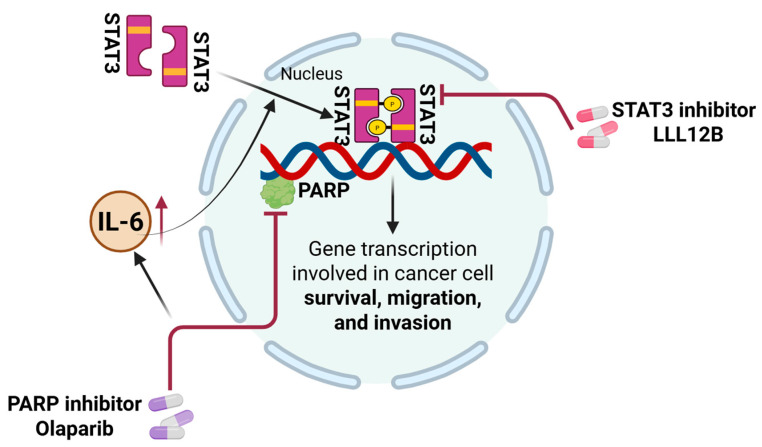
Schematic model illustrating the mechanism by which combination therapy with LLL12B enhances therapeutic efficacy in TNBC cells. Olaparib treatment upregulates IL-6 secretion in TNBC cells, leading to activation of the STAT3 signaling pathway. Activated STAT3 translocates to the nucleus, where it promotes the transcription of downstream target genes involved in cancer cell survival, migration, and invasion. Inhibition of STAT3 activation by LLL12B sensitizes TNBC cells to olaparib, resulting in reduced viability, migration, and invasion. The combination therapy exhibits synergistic anti-tumor effects.

## Data Availability

The original contributions presented in this study are included in the article/Appendix A. Further inquiries can be directed to the corresponding author.

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
