# Peer review of "Co-Inhibition of PARP and STAT3 as a Promising Approach for Triple-Negative Breast Cancer"

_biomolecules, 2025, doi:10.3390/biom15071035_

Round 1
Reviewer 1 Report
Comments and Suggestions for Authors
Triple-negative breast cancer (TNBC) is an aggressive and difficult-to-treat cancer. While the PARP inhibitor olaparib benefits some patients with BRCA mutations, it is not effective for all. This study explored combining olaparib with a STAT3 inhibitor, LLL12B, in both BRCA-mutant and non-mutant TNBC cells. The combination more effectively reduced cell growth, movement, and invasion than either drug alone. Blocking STAT3 also increased sensitivity to olaparib. The authors suggest that dual inhibition of PARP and STAT3 could be a promising treatment for TNBC, regardless of BRCA status. However, the findings also raise questions that require further clarification.
# It would be helpful if the authors could provide the number of cells used for siRNA transfection experiments, along with the concentrations of the siRNAs used.
# Figure 1: The rationale for using different doses of olaparib across cell lines when measuring IL-6 levels is unclear. How were these doses determined?
# The authors observed increased IL-6 levels in TNBC cells following olaparib treatment (Figure 1). Were p-STAT3 levels assessed in the same experimental context to support a mechanistic link?
# Given that activated STAT3 translocates to the nucleus to regulate target gene expression, did the authors examine p-STAT3 levels specifically in the nuclear fraction?
#Figure 2: What was the basis for selecting 15 µM olaparib to assess cell viability following STAT3 knockdown?
#Figure 3: The authors used varying doses of olaparib, LLL12B, and their combination across different TNBC cell lines. What criteria were used to determine these concentrations? Similar clarification is also needed for the dose selections in Figures 4 and 5.
#Do the authors offer any insights into the potential mechanisms underlying the observed synergistic effects of combination therapy on TNBC cell migration and invasion? They might consider including that in the discussion section.
#Please spell out the full forms of SSBs and DSBs in Lines 272–273 for clarity.
# It appears that the original images provided by the authors are mostly cropped western blots.
Author Response
# It would be helpful if the authors could provide the number of cells used for siRNA transfection experiments, along with the concentrations of the siRNAs used.
Response: Thank you for your valuable comment. We have updated the methods section to include additional details regarding the siRNA transfection experiments (page 4, line 146-147). Specifically, cells were transfected at approximately 60%–80% confluency using 100 nM of siSTAT3 or control siRNA.
# Figure 1: The rationale for using different doses of olaparib across cell lines when measuring IL-6 levels is unclear. How were these doses determined?
Response: We thank the reviewer for this important comment. The different doses of olaparib used across TNBC cell lines were chosen based on their variable sensitivity to the drug, as observed in our preliminary dose–response viability assays. To ensure a consistent biological context for comparing IL-6 induction, we selected olaparib concentrations that produced comparable levels of partial inhibition (approximately 30–50% reduction in cell viability) in each cell line. This strategy allowed us to evaluate IL-6 upregulation under equivalent sub-cytotoxic stress, minimizing confounding effects from excessive cell death. For example, SUM149 cells are very sensitive to olaparib—using 15 µM of olaparib would result in significant cell death.
We have clarified this rationale and added details to the methods section in the revised manuscript (page 3, line 132-137).
# The authors observed increased IL-6 levels in TNBC cells following olaparib treatment (Figure 1). Were p-STAT3 levels assessed in the same experimental context to support a mechanistic link?
Response: We appreciate the reviewer’s insightful comment. To investigate whether IL-6 induction by olaparib leads to activation of downstream signaling, we performed western blot analysis to assess phosphorylated STAT3 (P-STAT3) levels in TNBC cells treated with LLL12B and/or olaparib, or with DMSO alone. We observed increased P-STAT3 expressions following olaparib treatment compared to DMSO, supporting a mechanistic link between IL-6 upregulation and STAT3 pathway activation. These results are now included in Figure 4 and described in the revised Results section 3.4 (page8, line 290-294).
# Given that activated STAT3 translocates to the nucleus to regulate target gene expression, did the authors examine p-STAT3 levels specifically in the nuclear fraction?
Response: We thank the reviewer for this insightful comment. As suggested, we examined the nuclear localization of STAT3 in TNBC cells following olaparib treatment. Immunofluorescence (IF) staining revealed increased nuclear accumulation of STAT3 in olaparib-treated cells, as shown in Figure 4 and described in Results section 3.4. (page8, line 294-299).
#Figure 2: What was the basis for selecting 15 µM olaparib to assess cell viability following STAT3 knockdown?
Response: We thank the reviewer for this important question. The selection of 15 µM olaparib in the STAT3 knockdown experiment was based on a different experimental context from the IL-6 induction assay, which used a range of 5–30 µM. In the STAT3 knockdown study, we aimed to evaluate the effect of STAT3 silencing on olaparib sensitivity across multiple TNBC cell lines using a consistent drug concentration for direct comparison. We selected 15 µM as a middle-range dose that could induce measurable effects. As shown in Figure 2, despite using the same concentration, the extent of viability inhibition varied across the different cell lines, reflecting intrinsic differences in drug response.
#Figure 3: The authors used varying doses of olaparib, LLL12B, and their combination across different TNBC cell lines. What criteria were used to determine these concentrations? Similar clarification is also needed for the dose selections in Figures 4 and 5.
Response: We thank the reviewer for this important comment. In this study, the monotherapy effects of olaparib and LLL12B were first established and compared to their combination across multiple experimental assays, including MTT, migration, and invasion assays. The doses of olaparib, LLL12B, and their combination varied across these assays because the sensitivity of TNBC cells to these drugs can be influenced by several factors such as seeding cell density, cell growth rate, and the type and areas of experimental culture system used (e.g., 96-well plates for viability assays, 10-cm plates for protein analyses, and transwell chambers for invasion assays). Therefore, drug concentrations were optimized for each assay to achieve measurable biological effects while minimizing cytotoxicity that could complicate the interpretation of results. We have clarified this rationale in the revised manuscript in Materials and Methods section 2.1 and the legends of Figures 3, 4, and 5.
#Do the authors offer any insights into the potential mechanisms underlying the observed synergistic effects of combination therapy on TNBC cell migration and invasion? They might consider including that in the discussion section.
Response: We appreciate the reviewer’s insightful suggestion. Based on our results, the observed synergistic effects of the combination therapy (LLL12B + olaparib) on TNBC cell migration and invasion likely involve enhanced inhibition of the IL-6/STAT3 signaling pathway, which is known to regulate genes associated with tumor cell motility and invasiveness. Additionally, co-targeting these pathways may more effectively suppress downstream effectors involved in epithelial-to-mesenchymal transition (EMT) and matrix remodeling. We have expanded the Discussion section (Page 15, Line 436-443) in the revised manuscript to include these potential mechanisms and relevant literature to provide a more comprehensive interpretation of our findings.
#Please spell out the full forms of SSBs and DSBs in Lines 272–273 for clarity.
Response: Thank you for your comment. The full forms of SSBs and DSBs have been provided in the Introduction section for clarity. Please see the revised manuscript on page 1-2, line 40-42.
# It appears that the original images provided by the authors are mostly cropped western blots.
Response: Thank you for your comment. We would like to clarify that in our experiments we typically detect both the target protein and GAPDH from the same membrane. After the blocking step, the membrane is cut horizontally based on the molecular weights of the proteins of interest to allow for separate antibody incubation. The cropped western blot images shown in the main figures represent the relevant regions containing the target proteins in the same western blot membranes. The original blot images, which include the marked bands corresponding to the target proteins, have been provided in supplementary materials to ensure transparency and data integrity.
Reviewer 2 Report
Comments and Suggestions for Authors
I have reviewed the manuscript " Dual Inhibition of PARP and STAT3 as a Novel Therapeutic Strategy for Triple-Negative Breast Cancer". The authors present compelling in vitro evidence that targeting both PARP and STAT3 pathways using olaparib and LLL12B enhances therapeutic efficacy in both BRCA-mutant and BRCA-proficient triple-negative breast cancer (TNBC) cells. The study is scientifically relevant, well-structured, and explores a rational combination strategy for a cancer subtype with limited treatment options. However, the manuscript would benefit from more rigorous clarification of experimental details, correction of inconsistencies, and a deeper discussion of translational relevance. Line-by-line comments are provided below.
Line 20–22: The phrase "can induce increased interleukin-6 (IL-6) in TNBC cells" would benefit from a quantitative mention (e.g., 2–3 fold increase).
Line 28–29: Replace “regardless of BRCA mutation status” with “in both BRCA-mutant and BRCA-proficient TNBC cells” for precision.
Lines 47–49: Suggest rewording for clarity: "BRCA-proficient tumors retain homologous recombination capacity, reducing the efficacy of PARP inhibitors."
Line 66–70: This sentence describing compensatory mechanisms is useful—consider expanding briefly on what “alternative activation routes” means (e.g., cytokine signaling, non-canonical pathways).
Line 84: Suggest citing additional studies (2023–2024) that have evaluated direct STAT3 inhibitors beyond LLL12B for context.
Line 96: Specify the final concentrations of DMSO in all assays to reassure readers about control comparability.
Line 113: Clarify whether cells were authenticated and tested for mycoplasma contamination
Line 118: Include the IL-6 assay sensitivity (which is mentioned) and linear range—important for data interpretation.
Line 126: Was the siRNA-mediated STAT3 knockdown verified by densitometry or just visual inspection? Add details.
Line 150–157: For wound healing assay, indicate if serum-free conditions were maintained after scratch to exclude proliferation effects.
Line 168: Include the staining method used for cell invasion (e.g., crystal violet?)
Section 3.1 (Lines 176–189)
Line 180: The phrase “significantly induced” should be followed by a fold change or exact p-value.
Line 186: Clarify the order of increasing concentration (O5 → O30) in the figure legend.
Section 3.2 (Lines 190–208)
Line 199–201: Good observation—consider expanding in Discussion why STAT3 knockdown enhances olaparib response in both BRCA contexts (i.e., is STAT3 facilitating alternative DNA repair or survival signaling?).
Section 3.3 (Lines 209–225)
Line 210: There is a typo—“bazedoxifene” should likely be “LLL12B.” Correct throughout if this is an error.
Line 216: Consider running combination index (CI) or synergy scoring models (e.g., Bliss or Loewe) to formally confirm synergy.
Section 3.4 & 3.5 (Lines 226–260)
Line 233: Nice presentation—please specify how scratch closure or invasion % was calculated (ImageJ?).
Line 255–259: Report actual invasion inhibition values for key cell lines in the legend or text for stronger impact.
Lines 288–295: This section links IL-6 induction and STAT3’s role well. Consider noting whether IL-6/STAT3 activation also promotes DNA repair, which may explain resistance.
Lines 303–304: Consider citing other combinations of STAT3 inhibitors with PARP inhibitors in other cancers (e.g., ovarian, prostate).
Line 309: The conclusion about metastasis inhibition is promising. It would be stronger if the authors acknowledged the in vitro nature and suggested future in vivo validation.
Author Response
Line 20–22: The phrase "can induce increased interleukin-6 (IL-6) in TNBC cells" would benefit from a quantitative mention (e.g., 2–3 fold increase).
Response: Thank you for the suggestion. We have revised the text to specify the magnitude of IL-6 induction observed in our experiments (2- to 39-fold increase across 5 cell lines), based on ELISA quantification.
Line 28–29: Replace “regardless of BRCA mutation status” with “in both BRCA-mutant and BRCA-proficient TNBC cells” for precision.
Response: Revised as suggested to improve clarity and accuracy.
Lines 47–49: Suggest rewording for clarity: "BRCA-proficient tumors retain homologous recombination capacity, reducing the efficacy of PARP inhibitors."
Response: We agree and have revised the sentence as follows:
Importantly, most BRCA-proficient TNBCs show limited response to PARP inhibitor therapy due to their intact homologous recombination repair, which effectively counters PARP-induced DNA damage and reduces treatment efficacy.
Line 66–70: This sentence describing compensatory mechanisms is useful—consider expanding briefly on what “alternative activation routes” means (e.g., cytokine signaling, non-canonical pathways).
Response: Thanks for the comments, we have elaborated this alternative activation in the revised manuscript as follows:
Moreover, alternative routes such as cytokine and growth factor receptors, oncogenic kinases (e.g., SRC, BCR-ABL), or non-canonical pathways involving mitochondrial STAT3 and Ser727 phosphorylation can activate STAT3, allowing it to bypass upstream inhibition.
Line 84: Suggest citing additional studies (2023–2024) that have evaluated direct STAT3 inhibitors beyond LLL12B for context.
Response: Thank you for your suggestions. We have cited additional studies on direct STAT3 inhibitors in cancer as follows:
In addition to LLL12B, other direct STAT3 inhibitors such as TTI-101 (C188-9) [24], LYW-6 [25], and dual-site inhibitors like compound 4c [26] have shown promising antitumor effects in early-phase clinical or preclinical studies, further underscoring the therapeutic relevance of targeting STAT3 across cancer types.
Reference:
- Li, Y.; Dong, Y. TTI-101 targets STAT3/c-Myc signaling pathway to suppress cervical cancer progression: an integrated experimental and computational analysis. Cancer Cell Int 2024, 24, 286, doi:10.1186/s12935-024-03463-6.
- Wang, H.; Liu, Z.; Guan, L.; Li, J.; Chen, S.; Yu, W.; Lai, M. LYW-6, a novel cryptotanshinone derived STAT3 targeting inhibitor, suppresses colorectal cancer growth and metastasis. Pharmacol Res 2020, 153, 104661, doi:10.1016/j.phrs.2020.104661.
- Xu, S.; Fan, R.; Wang, L.; He, W.; Ge, H.; Chen, H.; Xu, W.; Zhang, J.; Xu, W.; Feng, Y.; et al. Synthesis and biological evaluation of celastrol derivatives as potent antitumor agents with STAT3 inhibition. J Enzyme Inhib Med Chem 2022, 37, 236-251, doi:10.1080/14756366.2021.2001805.
Line 96: Specify the final concentrations of DMSO in all assays to reassure readers about control comparability.
Response: Thank you for pointing this out. We have now specified the final concentrations of DMSO used in all relevant assays in the Methods section as follows:
In all assays, the final concentration of DMSO was kept consistent across experimental and control groups and did not exceed 0.1%, ensuring appropriate vehicle control comparability.
Line 113: Clarify whether cells were authenticated and tested for mycoplasma contamination.
Response: We appreciate your comment. The cell lines used in this study were authenticated by short tandem repeat (STR) profiling and were routinely tested for mycoplasma contamination using a PCR-based detection kit. All cell lines were confirmed to be mycoplasma-free prior to use in experiments. We added this clarity in the revised manuscript (page 3, line126-129).
Line 118: Include the IL-6 assay sensitivity (which is mentioned) and linear range—important for data interpretation.
Response: We thank the reviewer for this helpful suggestion. We have now included the sensitivity and linear detection range of the IL-6 ELISA assay in the manuscript (page4, line 141-142). Specifically, the assay has a sensitivity of 0.70 pg/mL and a linear detection range of 3.1–300 pg/mL, as reported by the manufacturer.
Line 126: Was the siRNA-mediated STAT3 knockdown verified by densitometry or just visual inspection? Add details.
Response: We thank the reviewer for this important suggestion. STAT3 knockdown efficiency was evaluated by Western blot analysis. We attempted densitometric quantification; however, in several cell lines (MDA-MB-436, SUM149, and MDA-MB-231), STAT3 bands were nearly or completely undetectable following siRNA treatment, making accurate quantification challenging. Nonetheless, the absence or drastic reduction of STAT3 signal supports robust knockdown. We have revised the methods section to include this clarification as follows:
Knockdown efficiency was evaluated by Western blot. Although densitometric quantification was attempted, in several cell lines (MDA-MB-436, SUM149, and MDA-MB-231), STAT3 protein levels were nearly or completely undetectable after STAT3 siRNA transfection in most of transfected TNBC cell lines, preventing us to do reliable quantifications. The absence or marked reduction of the STAT3 band indicated effective gene silencing.
Line 168: Include the staining method used for cell invasion (e.g., crystal violet?)
Response: We thank the reviewer for the helpful suggestion. We have now included the staining method used for the cell invasion assay in the Methods section (page5, line 216-218). Specifically, invaded cells were fixed with methanol and stained with 0.1% crystal violet before quantification.
Section 3.1 (Lines 176–189)
Line 180: The phrase “significantly induced” should be followed by a fold change or exact p-value.
Line 186: Clarify the order of increasing concentration (O5 → O30) in the figure legend.
Response: Thank you for your suggestions. We have revised the manuscript to include the fold changes for the IL-6 induction described in Line 180. Additionally, we clarified the order of olaparib concentrations (O5 → O30) in the figure legend of Figure 1 to indicate the increasing dose progression.
Section 3.2 (Lines 190–208)
Line 199–201: Good observation—consider expanding in Discussion why STAT3 knockdown enhances olaparib response in both BRCA contexts (i.e., is STAT3 facilitating alternative DNA repair or survival signaling?).
Response: We thank the reviewer for this insightful suggestion. We have expanded the discussion section (page14. line 410-415) to elaborate on the potential mechanisms by which STAT3 knockdown enhances olaparib sensitivity in both BRCA-mutant and BRCA-proficient TNBC cells. Specifically, we discuss how STAT3 may facilitate alternative survival signaling and influence DNA repair pathways, contributing to resistance against PARP inhibition. This dual targeting strategy may effectively disrupt these compensatory mechanisms, improving therapeutic efficacy regardless of BRCA status.
Section 3.3 (Lines 209–225)
Line 210: There is a typo—“bazedoxifene” should likely be “LLL12B.” Correct throughout if this is an error.
Response: Thank you for pointing out this error. We have corrected this and similar typos throughout the entire manuscript.
Line 216: Consider running combination index (CI) or synergy scoring models (e.g., Bliss or Loewe) to formally confirm synergy.
Response: We strongly agree with your suggestion. However, the co-author who performed this experiment is no longer working in our lab. The data in figure 3 did not have performed for at least two different concentrations for each drug, which is insufficient to calculate a combination index. However, single drug treatment and combination treatment showed statistically significant differences compared to DMSO control. In addition, combination treatments showed statistically significant compared to each single drug treatment (P<0.01 - P<0.0001).
Section 3.4 & 3.5 (Lines 226–260)
Line 233: Nice presentation—please specify how scratch closure or invasion % was calculated (ImageJ?).
Response: Thank you for your comment.
For the migration assay, the scratch area was measured using the Echo Rebel microscope (Echo, San Diego, CA, USA) equipped with integrated image capture and analysis software. Images were taken at 0 hours (prior to drug treatment) and after incubation. The scratch boundaries were traced using the polygon area measurement tool within the Echo software, which automatically calculates the area based on calibrated image dimensions. The scratch closure rate was then calculated using the formula: (initial scratch area – final scratch area) / initial scratch area, and the relative migration percentage was normalized to the DMSO-treated control.
For the invasion assay, following drug treatment and incubation, cells that invaded through the Matrigel-coated inserts were fixed, stained, and imaged. The number of invasive cells was counted using ImageJ, and results are presented as invasive cell numbers to compare drug-treated versus control groups. We have revised the method for these two experiments, see Method 2.8 and 2.9 in the revised manuscript.
Line 255–259: Report actual invasion inhibition values for key cell lines in the legend or text for stronger impact.
Response: Thank you for the helpful suggestion. We have now included the actual invasion inhibition values relative to the DMSO control for the TNBC cell lines (Figure 6F) in the Results section to enhance the clarity and impact of our findings.
Lines 288–295: This section links IL-6 induction and STAT3’s role well. Consider noting whether IL-6/STAT3 activation also promotes DNA repair, which may explain resistance.
Response: Thank you for your thoughtful comment. We have revised the text to address this important point. Specifically, we now note in the Discussion (Page 14, Line 410-415) that IL-6/STAT3 signaling has been implicated in promoting DNA damage repair pathways, including homologous recombination and non-homologous end joining, in various cancer types. This may contribute to resistance to DNA-damaging agents like olaparib.
Lines 303–304: Consider citing other combinations of STAT3 inhibitors with PARP inhibitors in other cancers (e.g., ovarian, prostate).
Response: Thank you for the insightful suggestion. We have cited other combinations of STAT3 inhibitors with PARP inhibitors in ovarian and ER+ breast cancers (page15, line 427-431). These additions support the broader rationale for targeting STAT3 to potentiate PARP inhibitor responses across multiple tumor types.
Line 309: The conclusion about metastasis inhibition is promising. It would be stronger if the authors acknowledged the in vitro nature and suggested future in vivo validation.
Response: We appreciate the reviewer’s positive feedback on our conclusion regarding metastasis inhibition. We agree that the current findings are based on in-vitro assays, which, while informative, do not fully replicate the complexity of the in-vivo tumor microenvironment. We have revised the manuscript to acknowledge this limitation and have added a statement suggesting that future in vivo studies are warranted to validate the anti-metastatic potential of the olaparib and LLL12B combination in TNBC models.
Round 2
Reviewer 1 Report
Comments and Suggestions for Authors
I appreciate the authors for thoroughly addressing all the concerns and revising the manuscript accordingly.
Reviewer 2 Report
Comments and Suggestions for Authors
I am satisfied with the authors’ responses to my previous comments and appreciate the revisions they have made to address the concerns raised. The manuscript is now significantly improved, and I believe it meets the standards for publication. I have no further concerns, and I recommend the paper for acceptance in its current form.